# Effects of Different Biostimulants on Growth and Development of Grapevine Seedlings under High-Temperature Stress

**Jiuyun Wu** [1,2,†], **Haixia Zhong** [1,2,†], **Yaning Ma** [1], **Shijian Bai** [1,3], **Vivek Yadav** [2], **Chuan Zhang** [1,2], **Fuchun Zhang** [1,2], **Wei Shi** [4], **Riziwangguli Abudureheman** [1,*] and **Xiping Wang** [1,5,*]

1   Turpan Research Institute of Agricultural Sciences, Xinjiang Academy of Agricultural Sciences, Xinjiang Grape Engineering Technology Research Center, Turpan 838000, China; kobewjy@163.com (J.W.); zhonghaixia1@sina.cn (H.Z.); 18599491754@163.com (Y.M.); bsj19861103@163.com (S.B.); 2016204005@njau.edu.cn (C.Z.); zhangfc@xaas.ac.cn (F.Z.)
2   The State Key Laboratory of Genetic Improvement and Germplasm Innovation of Crop Resistance in Arid Desert Regions (Preparation), Key Laboratory of Genome Research and Genetic Improvement of Xinjiang Characteristic Fruits and Vegetables, Institute of Horticultural Crops, Xinjiang Academy of Agricultural Sciences, Urumqi 830091, China; vivekyadav@nwafu.edu.cn
3   Xinjiang Uighur Autonomous Region of Grapes and Melons Research Institution, Shanshan 838200, China
4   Turpan Eremophytes Botanic Graden, Xinjiang Institute of Ecology and Geography, Chinese Academy of Sciences, Urumqi 830091, China; shiwei@ms.xjb.ac.cn
5   Colleges of Horticulture, Northwest A&F University, Yangling 712100, China
*   Correspondence: 13289951672@163.com (R.A.); wangxiping@nwsuaf.edu.cn (X.W.); Tel.: +86-29-87082129 (X.W.)
†   These authors contributed equally to this work.

**Abstract:** High temperatures significantly affect the growth and development of grapevines, cause irreversible damage to plants, and severely impact grape production and quality. Biostimulants can promote the growth of plants and enhance their resistance to adverse stress. However, the effects of biostimulants on grapevines under high temperatures have not been studied in detail. To analyze the effects of various biostimulants on the growth and development of grape seedlings under high temperatures, we measured chlorophyll fluorescence parameters with observed seedling pheno-types under high temperatures in open field conditions in Turpan. We conducted a comprehensive analysis of the effects of different biostimulants on the growth, development, and photosynthesis of grapevine seedlings. Our study aimed to provide scientific evidence to improve cultivation methods for grapevines under high-temperature stress. The results revealed that biostimulants have a positive effect on promoting the growth of grapevine seedlings under high-temperature stress conditions. They also positively affect the accumulation of chlorophyll components in grapevine leaves, in-hibiting chlorophyll degradation and maintaining photosynthesis. However, the effects of different biostimulants were inconsistent. A comprehensive analysis revealed the following effectiveness order: T2 > T1 > T3 > Control. These findings suggest that T2 is the most effective in alleviating high-temperature stress and promoting grapevine growth. We recommend the use of T2 to improve the cultivation of grapevine seedlings during high-temperature periods. This has implications for grape production in hot and arid climatic areas.

**Keywords:** grapevine; heat stress; biostimulant; JIP-test





## 1. Introduction

Grapevines (*Vitis vinifera* L.) are sessile organisms that cannot change their location in the field and inevitably face various biological and abiotic stresses during their growth and development. The grape is an economically important crop species worldwide, but its quality and production are often limited by high-temperature stress. With global warming, high temperatures have become one of the primary abiotic stress factors that restrict the yield and quality of grapes [1–3]. Turpan, one of the most important grape

production areas in Xinjiang, China, covers an area of 36,253 hectares with a yield of 1,447,800 tons. 'Thompson Seedless' is the main cultivated grape variety in Turpan, with a planting area of 32,640 hectares (https://www.tlf.gov.cn/tlfs/sjkf/tlfdata.shtml, accessed on 28 November 2023). However, due to the distinctive geography of the Turpan region, its temperatures always exceed 40 °C for more than 35 days each summer. This intense heat can cause grapevines to lose water, wilt, and suffer cell damage. It can also disrupt their photosynthesis process. This seriously affects the growth and development of grapevine seedlings and causes irreversible damage to plants, resulting in stunted growth and even death of the plants [4–6].

Biostimulants are agronomic products that have become highly important in agriculture because they are formulated with substances capable of stimulating physiological and biochemical processes in plants, which help them adapt to different detrimental environmental conditions [7]. In general, a biostimulant is an agrochemical product formulated with mixtures of natural substances or microorganisms, which is used for enhancing nutrition efficiency and crop quality traits [8]. Moreover, biostimulants have varying effects on different plant species or cultivars, supporting the activity of microorganisms and serving as substrates for the formation of biologically active substances by these microorganisms. They can enhance the tolerance to stress in plants, and it has been determined that using biostimulants can improve the efficiency of mineral nutrition, resistance to abiotic stress (drought, high temperatures, salinity, heavy metals, etc.), and crop yield, or enhance quality characteristics, regardless of its content in essential mineral nutrients for plants, to maintain a good agronomic yield and quality of harvest under these conditions [9]. Furthermore, recent studies have evidenced the important role of biostimulants in minimizing cellular oxidative stress in plants. For instance, plant-based modified biostimulants (copper chlorophyllin) were used for plants under salinity stress, and findings showed that cellular oxidative stress was decreased in Arabidopsis thaliana plants [10]. A recent study also showed that seaweed extract biostimulants affected the sugarcane morphology and physiology through significant changes in oxidative stress [11].

Photosynthesis is one of the most sensitive physiological processes highly affected by temperature conditions. Leaves, being crucial organs for plant photosynthesis and gas exchange, exhibit the most obvious signs in response to heat stress [10]. High temperatures not only damage the tissue structure of grapevine leaves but also inhibit photosynthesis and nutrient metabolism. Additionally, it affects the growth and development of grapevine plants, leading to an imbalance in energy metabolism and material transformation processes. Eventually, this can also result in heat damage, malaise, or even the death of the plant [12–14]. A number of studies have reported that biostimulants have a significant impact on the growth and development of crops such as Capsicum, Malus, Nicotiana, and others. Additionally, the use of biostimulants can alleviate damage caused by heat stress. Biostimulants are widely used in crop cultivation due to their high efficiency, environmental friendliness, and absence of residues [15–19]. However, the effects of biostimulants on grapevines under high-temperature stress remain poorly understood. Therefore, we conducted research on the effects of different biostimulants on the growth and development of grapevine seedlings under natural high temperatures. Our aim is to provide a basis for reinforcing the cultivation of grape seedlings, offering a reference for alleviating heat stress, and exploring pathways for stress-resistant cultivation.

## 2. Materials and Methods

### 2.1. Plant Material and Treatments

'Thompson Seedless' (*V. vinifera*) is the main cultivated variety in Turpan grape production area, which covers a planting area of 32,640 hectares. We selected 'Thompson Seedless' (*V. vinifera*) grape plants as an experimental material and cultivated them in plastic pots at the grapevine garden of the Turpan Research Institute of Agricultural Sciences, Xinjiang Academy of Agricultural Sciences (XAAS). The garden is located at 89°11′ E, 42°56′ N, at an altitude of 0 m [10]. Uniform grapevine seedlings aged one year (1a) were

selected and transplanted into pots. For the experiment, 1-year-old grapevine seedlings with almost the same height and stem diameter were cultivated in plastic pots containing soil mixed uniformly with substrate, nutrient soil, and field soil in a 1:1:1 (*v*/*v*/*v*) ratio. The plastic pots, with dimensions of 37.5 cm in diameter and 40.5 cm in height, were used for the cultivation. To ensure the optimal growth of the seedlings during observation and sampling, all containers were mixed with the same soil uniformly, and the soil moisture was carefully maintained at a moderate level.

Three treatments were planned, and the grapevine seedlings were treated with biostimulants through a combination of foliar spraying and root irrigation. The plants were sprayed and irrigated approximately once every 7 days. Water was used as a control (C), following the same cultivation conditions as the test treatments. Each treatment is sprayed with a watering can (200 mL) filled with biostimulants when there is no wind and the air temperature is below 30 °C (9:30–10:30). The upper and lower leaves are wet as is the standard to ensure all leaves are evenly sprayed. The plants are fully irrigated by mixing every 100 mL of biological agent with 1000 mL of water. Each experiment utilized every single plant in the container, with each treatment having three grapevine seedlings and three biological replicates. The biostimulant products with patents were researched and provided by Turpan Eremophytes Botanic Garden, Xinjiang Institute of Ecology and Geography, Chinese Academy of Sciences. The biostimulants were named T1 (Main compounds: β-Myrcene; Patent No.: ZL202010410895.1), T2 (Main compounds: BaZFP924 protein; Patent No.: ZL202110900671.3), and T3 (Main compounds: Aspergone; Patent No.: ZL202210880877.9), respectively.

### 2.2. Temperature Measurement Tested

Throughout the period of high-temperature conditions in the grape research farm, we employed the MicroLite USB Temperature Data Logger (manufactured by Fourier Systems, Fourtec-Fourier Technologies, Ltd., San Francisco, CA, USA) to monitor temperature fluctuations at a specific location. Temperature readings were recorded and logged every hour, covering the entirety of the high-temperature span from 1 May to 31 August 2022.

### 2.3. Plant, Leaf and Root Observation and Tested

The height of the plants, the length of the internodes (3rd, 4th, 5th, 7th, and 9th internodes in the branch of the vine), the stem diameter, and the root length (cm) were measured using a ruler. Additionally, the dry weight and fresh weight of the roots (g) were determined using an electronic balance. Three healthy, fully opened leaves from the 5th to 9th positions from the top were randomly selected and labeled on each plant. Chlorophyll (Chl) levels were measured using a TYS-B tester (Zhejiang Tuopu Yunnong Technology, Ltd., Hangzhou, Zhejiang, China) from 9:00 to 11:00, expressed by SPAD. The leaves were wiped clean before determination, and measurements were taken every 10 days.

For each grapevine seedling, three leaves were selected from 5th to 9th nodes from the top. The leaf area ($mm^2$) was measured using a leaf area meter. Subsequently, the fresh leaf mass (g) was determined using an electronic scale. The leaves were then placed in a room for drying, and the dry leaf mass (g) was measured after complete drying.

### 2.4. Measurement of Polyphasic Chl Fluorescence Transient OJIP

We selected fully opened grapevine leaves from the 5th to 9th positions from the top at 12:00 to 16:00 on a sunny day with no wind or clouds for measuring Chl, a fluorescence parameter, using the OJIP test. The parameters were measured using the Fluor Pen FP110 (Ecotech Ecological Technology Co., Ltd., Drásov, Czech Republic). Before measurements, leaves were acclimated to darkness for 20 min; the parameters included $F_0$, $F_v$, $F_m$, $F_v/F_0$, $F_v/F_m$, $\Psi_0$, $\varphi E_0$, $PI_{ABS}$, $ABS/RC$, $TR_0/RC$, $ET_0/RC$, $DI_0/RC$, etc. to provide information on the photochemical activity of PSII and the status of the PQ-pool [20].

*2.5. Data Analysis*

The test data were used for variance analysis, employing the LSD method for multiple comparisons and assessing the significance of differences. A significance level of $p < 0.05$ was considered for differences, while $p < 0.01$ denoted an exceptionally significant distinction. ANOVA, PCA, and figure generation were conducted using GraphPad Prism ver.9.0 (October 2020, Dotmatics Corporation, Boston, MA, USA).

## 3. Results

*3.1. Temperature Dynamics in the Field*

In May, the average temperature in the field was 30.52 °C, with the maximum temperature reaching 42.65 °C. By late June, the region had transitioned into a high-temperature phase, experiencing a peak of 44.29 °C, featuring 26 days surpassing 35 °C and 12 days exceeding 40 °C. July witnessed a continuous temperature rise, registering an average temperature of 32.92 °C and a maximum of 44.29 °C. During this period, 30 days surpassed 35 °C, with 14 days exceeding 40 °C. August marked the onset of an extremely high-temperature period, reaching a maximum of 45.67 °C. This scorching phase included 26 days beyond 35 °C and a notable 18 days surpassing 40 °C (refer to Table 1). Throughout the recorded high-temperature period spanning June to August, the maximum temperature soared to 45.67 °C, while the lowest dipped to 18.90 °C, resulting in an average temperature of 32.67 °C. Turpan experienced the initiation of the high-temperature period in early June 2022 (Figure 1), culminating in an extreme high-temperature phase in August 2022. The average temperature during this period was 32.13 °C, with a maximum temperature soaring to a remarkable 45.67 °C.

**Table 1.** Temperature variation during high-temperature period (2022).

| Air Temperature | May | June | July | August |
|---|---|---|---|---|
| Maximum Temperature/°C | 42.65 | 44.29 | 44.28 | 45.67 |
| Minimum Temperature/°C | 16.82 | 22.27 | 22.10 | 18.90 |
| Average Temperature/°C | 30.52 | 32.92 | 33.62 | 31.49 |
| ≥35.00 °C/Day | 27.00 | 26.00 | 30.00 | 26.00 |
| ≥40.00 °C/Day | 9.00 | 12.00 | 14.00 | 18.00 |

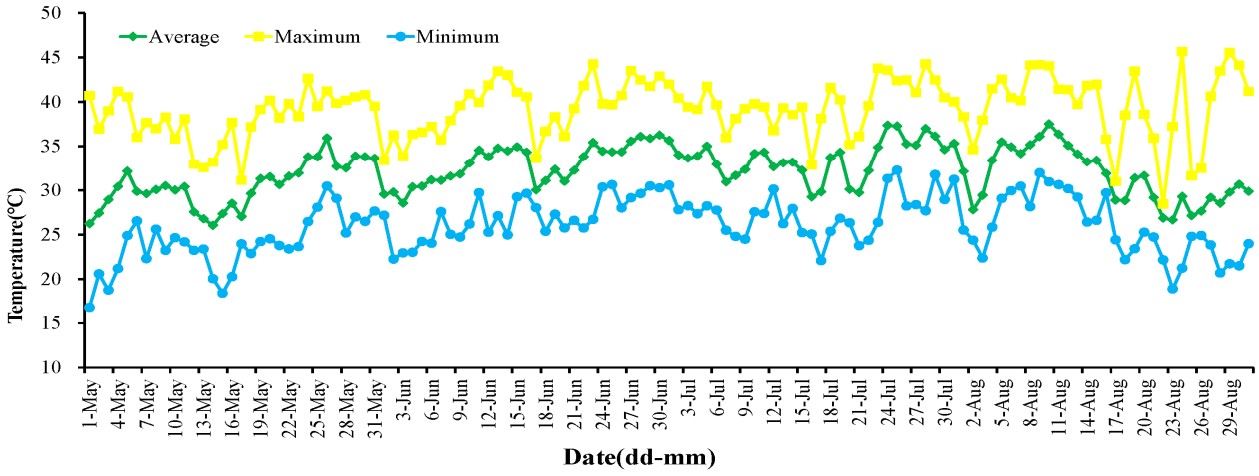

**Figure 1.** The temperature of the viticultural region of XAAS in summer in Turpan.

*3.2. Plants and Roots Distribution*

During the plant growth and development stage (30-May to 12-September), the control group always exhibited the smallest plant type and the shortest plant height, which was significantly lower than that of the treated groups ($p < 0.05$, 12-September). The root distri-

bution in the control group was relatively smaller, with the shortest root length, indicating weaker overall plant growth compared to the treated groups (see Figures 2 and 3a).

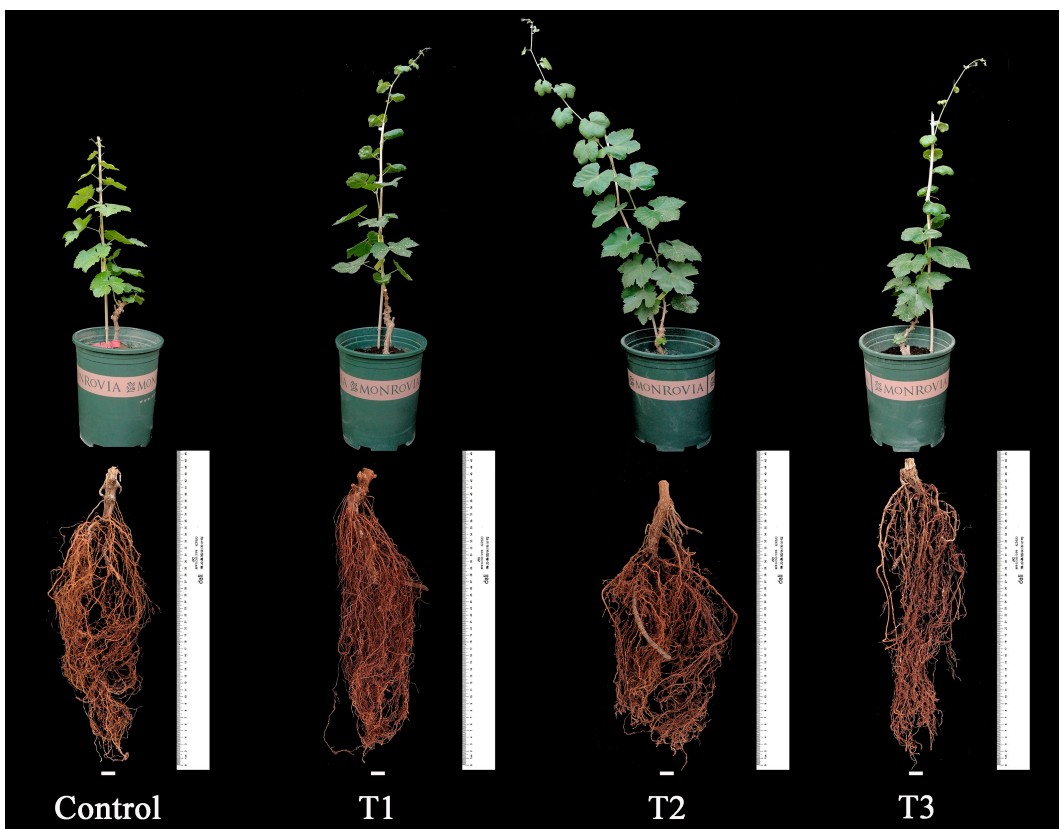

**Figure 2.** Effects of different biostimulants on plants and roots of grapevine.

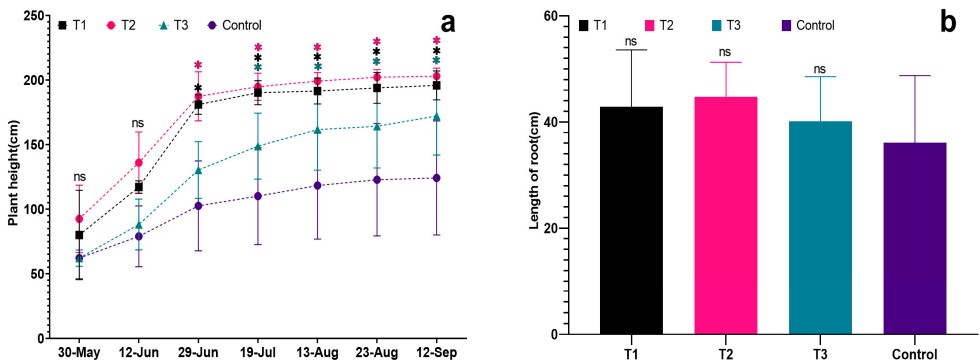

**Figure 3.** Effects of different biostimulants on plant height and root length of grapevine: (**a**) Plant height of grapevine; (**b**) Root length of grapevine. Graphs are plotted with the mean ± SE of three replications. * represent significant differences in expression levels at $p < 0.05$ and $p < 0.01$. "ns" is used to show non-significant differences.

Among the treated groups, T2 showed significantly better plant growth than the other treatments, and the seedlings' growth followed the order: T2 > T1 > T3 > Control. The results revealed a trend of fast growth in the early stage and slow growth in the later stage for grapevine seedlings. The average plant height for each treatment was 195.90 cm, 203.00 cm, and 172.20 cm, respectively. The control group had the shortest average plant height (only 123.20 cm), significantly lower than the treated groups ($p < 0.05$). Additionally, the control group exhibited a relatively smaller root distribution, the shortest root length, and weaker overall plant growth compared to the treated groups. In contrast,

T2 had an average plant height of 203.00 cm, significantly higher than the other treated groups ($p < 0.05$). T2 plants demonstrated the greatest strength, with more extensive root distribution, better growth of fresh branches, and overall plant development (see Figures 2 and 3a).

The root length of each treatment was significantly greater than that of the control (Figure 3b), with the root lengths ranked as follows: T2 > T1 > T3 > Control. The respective root lengths were 42.87 cm, 44.73 cm, 40.17 cm, and 36.10 cm. Notably, T2 exhibited the longest root length at 44.73 cm. The fresh root weight of T1 was the highest (66.0 g), followed by T2 (51.0 g) and T3 (28.3 g). In comparison, the control group had the lowest fresh root weight (24.7 g), significantly lower than each treatment ($p < 0.05$) and notably lower than T1 and T2 ($p < 0.01$). This indicates that all three biostimulants had a positive impact on the root growth of grapevine seedlings. Examining dry root weight, T2 had the highest value (17.7 g), followed by T3 (16.0 g) and T1 (15.0 g). The control group exhibited the lowest dry root weight (13.7 g), significantly lower than T2 ($p < 0.01$). The order of dry root weight was T2 > T1 > T3 > Control, with the dry root weights of each treatment being significantly higher than that of the control ($p < 0.05$).

### 3.3. Growth of Grapevine Seedling

All grapevine seedlings exhibited certain differences in height during the initial measurement on 30 May (see Figure 3a), These height growth for each treatment was 37.17 cm, 36.83 cm, 30.40 cm, and 38.83 cm, respectively, although none of these differences were deemed significant ($p < 0.05$). However, vines began to grow rapidly in June, the plant height has grown in 22.93 cm to 63.83 cm, height of T1 increased is the greatest, control is the lowest, was significantly lower than other treatments (see Table 2). By the last measurement on 12 September, the plant heights for each treatment were recorded as 195.90 cm, 203.00 cm, 172.20 cm, and 123.20 cm for the control group. These values represented a notable increase compared to the initial measurements on 30 May. The recorded growth for each treatment was 115.90 cm, 103.80 cm, and 114.40 cm, respectively. In contrast, the control group only exhibited a growth of 83.00 cm, which was significantly lower than that of T1 ($p < 0.01$). Upon analyzing the internode length of different nodes, all grapevine seedlings showed certain differences in internode length during the initial measurement on 30 May (see Figure 4b). Although there were no significant differences, it was observed that the internode length of each node in the control group was lower than that of the treated groups. Furthermore, the internode length of T1 was the maximum, consistently surpassing the lengths observed in other treatments, which is significantly greater than the control in the seventh and ninth internodes. This indicates that T1 had a more pronounced effect on the height growth of grapevine seedlings (Figure 4b).

**Table 2.** Effects of different biostimulants on grapevine growth.

| Group | T1 | T2 | T3 | Control |
|---|---|---|---|---|
| 30-May | 37.17 ± 37.59 [aA] | 36.83 ± 5.11 [aA] | 30.40 ± 33.19 [aA] | 38.83 ± 32.37 [aA] |
| 12-June | 63.83 ± 10.49 [aA] | 51.40 ± 12.61 [abA] | 42.17 ± 16.00 [abA] | 22.93 ± 21.03 [bA] |
| 29-June | 9.17 ± 4.37 [aA] | 7.43 ± 8.79 [aA] | 18.50 ± 22.15 [aA] | 7.23 ± 12.36 [aA] |
| 13-August | 1.33 ± 1.04 [aA] | 4.33 ± 4.01 [aA] | 12.83 ± 16.64 [aA] | 8.00 ± 7.76 [aA] |
| 23-August | 2.43 ± 2.53 [aA] | 3.00 ± 1.00 [aA] | 2.50 ± 2.00 [aA] | 4.50 ± 3.60 [aA] |
| 12-September | 1.93 ± 1.44 [aA] | 0.83 ± 0.58 [bAB] | 8.00 ± 4.82 [bAB] | 1.50 ± 0.87 [bB] |

The average values are means ± S.E. Different lowercase letters in each column indicate a significant difference at $p < 0.05$; Uppercase letters in each column indicate a significant difference at $p < 0.01$.

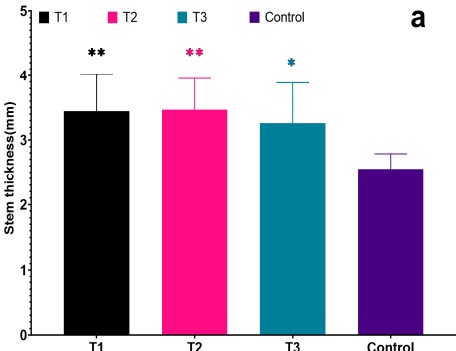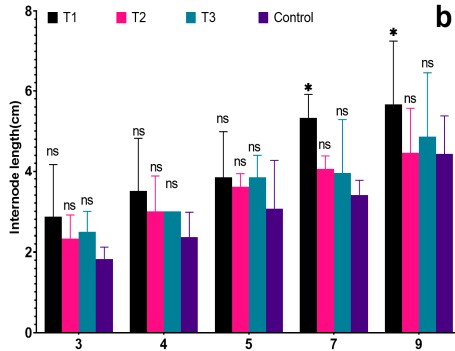

**Figure 4.** Effects of different biostimulants on stem diameter and internode length of grapevine: (**a**) Stem thickness; (**b**) Internode length. Graphs are plotted with the mean ± SE of three replications. * and ** represent significant differences in expression levels at $p < 0.05$ and $p < 0.01$. "ns" is used to show non-significant differences among the treatments.

In addition, the stem diameters of the seedlings in each treatment were 3.45 cm, 3.47 cm, 3.26 cm, and 2.55 cm, respectively. The stem diameters of the seedlings in each treatment were significantly greater than the control ($p < 0.05$). Moreover, the stem diameters of T1 were significantly greater than those of the control ($p < 0.01$). This suggests that all of the biostimulants had a certain effect on promoting the shoot growth of grapevine seedlings. Notably, T2 had a more pronounced effect on the stem diameter of grapevine seedlings (Figure 4a).

### 3.4. Leaves and Chl Content

In Figure 5a,b, it is evident that the dry and fresh leaf weights of grapevine seedlings in each treatment exceeded those of the control. Notably, T2 exhibited the highest fresh leaf weight at 1.10 g, followed by T1 (1.03 g) and T3 (0.96 g). In comparison, the control group had the smallest average fresh leaf weight at 0.95 g, which was obviously lower than T2 ($p < 0.05$). In leaf dry weight, T2 had the highest value (0.31 g), followed by T1 and T3 at 0.29 g and 0.28 g, respectively. The control had the lowest dry leaf weight, measuring 0.27 g, which was significantly lower than T2 ($p < 0.05$). In terms of leaf area, the treated plants showed a larger leaf area compared to the control group (see Figure 5c). The ranking of leaf area for the treatments was as follows: T1 > T2 > T3 > Control. Specifically, the leaf areas were 95.76 cm$^2$ for T1, 90.61 cm$^2$ for T2, 84.83 cm$^2$ for T3, and 79.38 cm$^2$ for the control. There was a significant difference in leaf area between T1 and the control, with T1 having the largest leaf area and the control having the smallest ($p < 0.05$). The chlorophyll (Chl) levels directly reflected the nutritional and photosynthetic capacity of plants. In each treatment, there was an initial increase in Chl, followed by a decrease and a slight rise again. Initially (on 30 May), the Chl levels were almost the same for all treatments. However, as the plants grew and developed, and with temperature changes, the Chl levels in the leaves began to vary. Notably, T2 consistently showed the highest Chl levels for most of the observation period. In the early stages, the control had slightly higher Chl levels than T2 and T3, but as time progressed, it became lower than each treatment, particularly during the middle of the high-temperature period (after 19 July). In particular, the Chl content of T3 was always lower than other treatments in the early stage, but it was significantly higher than others treated in the later stage, indicating that all three biostimulants had a positive effect on the Chl accumulation of grapevine leaves or alleviating the degradation of Chl, and the Chl of grapevine leaves was always maintained at a high level in treatment, which was facilitating the growth of grapevine seedlings under the high temperature.

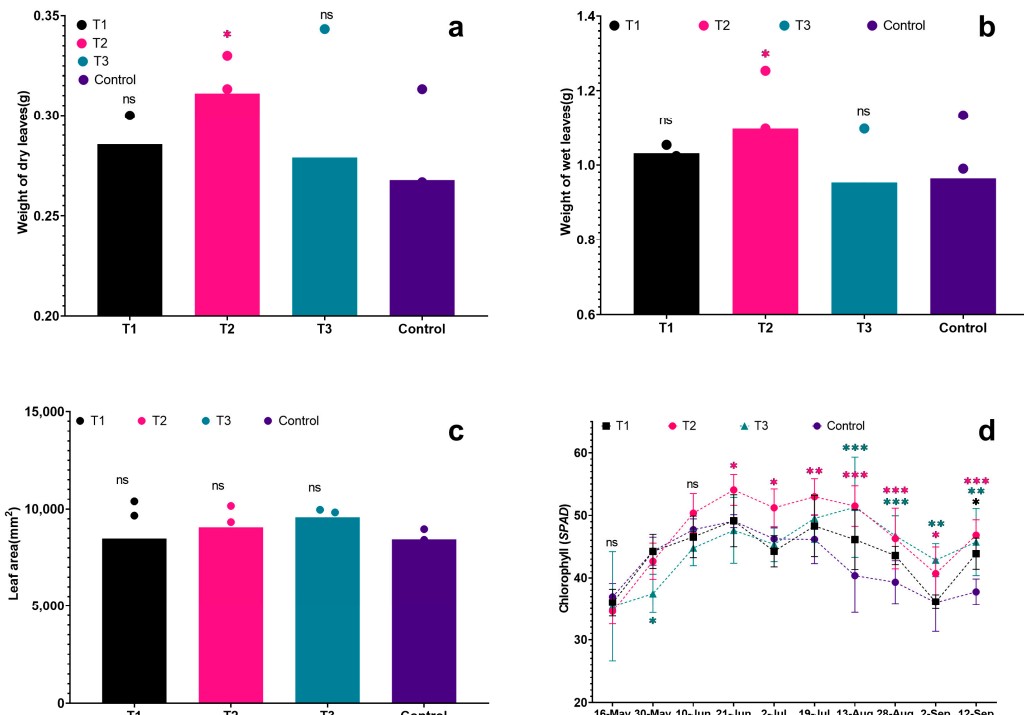

**Figure 5.** Effects of different biostimulants on grape leaves: (**a**) Weight of dry leaves; (**b**) Weight of fresh leaves; (**c**) Leaf area; (**d**) Chl content. Colour represent treatment. Graphs are plotted with the mean ± SE of three replications. *, ** and *** represent significant differences in expression levels at $p < 0.05$, $p < 0.01$ and $p < 0.001$. "ns" is used for non-significant differences.

### 3.5. Polyphasic Chl Fluorescence Transient OJIP

There was no significant difference between the treatments in the OJIP normalization curve from June to August (Figure 6a,c,e). However, it is evident that the OJIP curve of the control was slightly higher than that of the treated group in June (Figure 6a). Results of the high-temperature period (July and August) showed that the OJIP curve of the control became significantly lower than that of each treated group (Figure 6c,e). Therefore, Through the double normalization of the OJIP curve data, it becomes evident that the O-J phase, J-I phase, and I-P phase of each treatment were significantly lower than those of the control, except for the time in the I-P phase of T1 (Figure 6b). This suggests that the photosynthetic efficiency of the control was significantly higher than that of the other treatments in the high-temperature early stage (June).

However, during the high-temperature period in July, except for T1, which was slightly lower than the control in the O-J phase for a brief period, the photosynthetic efficiency of the three treatments in the O-J and J-I phases was consistently higher than that of the control. This indicates that the photosynthetic efficiency of the seedlings in the three treatments at this stage was significantly higher than that of the control (Figure 6d). After the high-temperature period in August, both T2 and T1 exhibited significantly higher values during the O-J phase and J-I phase. However, in the I-P phase, they were lower than the control values. Specifically, T1 was significantly lower than the control in the O-J phase, J-I phase, and I-P phase, as illustrated in Figure 6f.

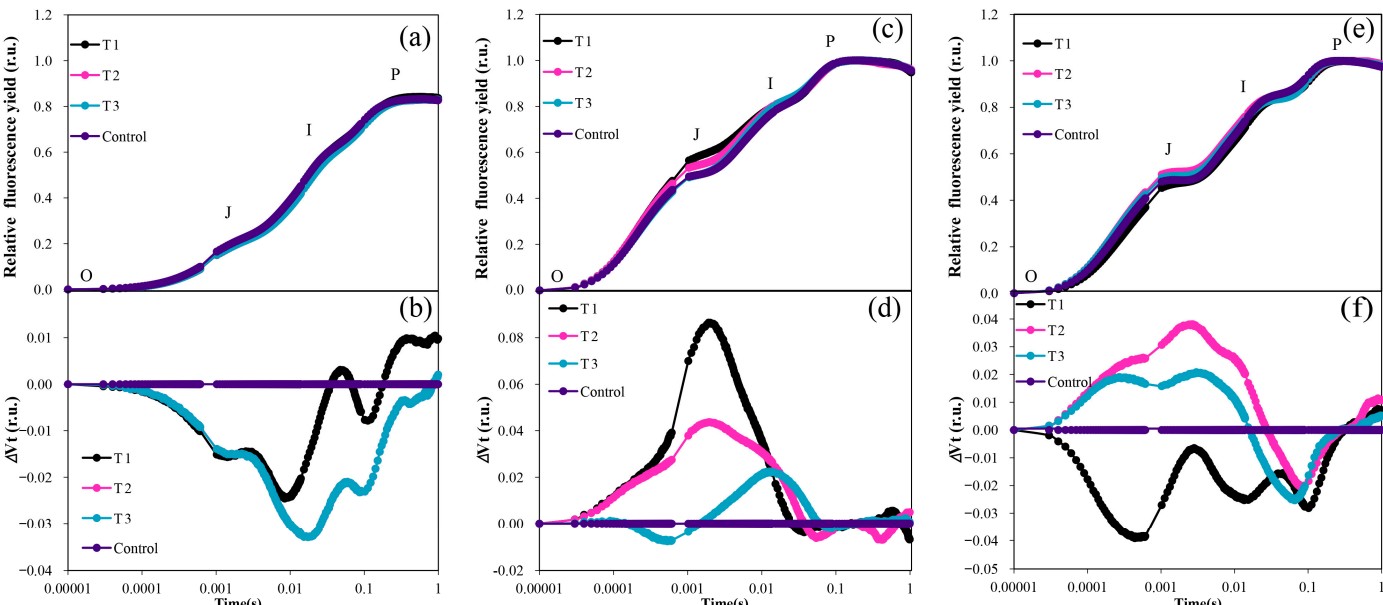

**Figure 6.** OJIP curves of grapevine for different treatments under high−temperature conditions: (**a**) Normalized OJIP curve for June; (**b**) Double normalized OJIP curve for June month; (**c**) Normalized OJIP curve for July (**d**) Double normalized OJIP curve for July month; (**e**) Normalized OJIP curve for August; (**f**) Double normalized OJIP curve for August.

### 3.6. Chl a Fluorescence Parameters

$F_0$ reflected the degree of damage to the thylakoid membrane. A higher $F_0$ value indicates more severe damage to the thylakoid membrane. The maximum fluorescence $F_m$ reflects the electron transport from Photosystem II (PSII), and a lower Fm value indicates a higher degree of heat damage. The $F_v/F_0$ represents the potential activity of PSII, reflecting the activity of the PSII center. The maximum photochemical quantum yield, $F_v/F_m$, characterizes the photoenergy conversion efficiency of the PSII center. $\Psi_0$ represented the ratio of electron transport optical quantum flux in the captured light quantum flux. $\varphi E_0$ represents the photoquantum yield for electron transport, reflecting the proportion of absorbed light quanta that transport electrons to other downstream electron acceptors. $PI_{ABS}$ is the photochemical performance index based on absorbed light energy and $ABS/RC$ is the absorbed light quantum flux at the reaction center of PSII. $TR_0/RC$ is the initial (or maximum) captured light quantum flux of the reaction center, $ET_0/RC$ is the initial electron transport light quantum flux of the reaction center, and $DI_0/RC$ reflects the ratio of energy dissipated by the PSII reaction center as thermal energy.

The results revealed that the control group exhibited lower values compared to the treated groups in terms of $F_v/F_m$, $F_v/F_0$, $F_v$, $\varphi E_0$, $PI_{ABS}$, and other fluorescence parameters. Additionally, $DI_0/RC$ was significantly higher in the control group than in each treated group. This indicates that during the high-temperature period (August), the light energy conversion efficiency, activity of the PSII center, optical quantum yield, and photochemical properties of the control group were lower than those of each treated group. Furthermore, the heat dissipation ratios in the control group were significantly higher than in all the treated groups. These findings suggest that all three biostimulants had a certain effect on the photosynthetic efficiency of grapevine seedlings under high-temperature conditions. T2 was significantly higher than that of others treated in aspects of $\Psi_0$, $F_v/F_m$, $F_v/F_0$, $\varphi E_0$, and $PI_{ABS}$ (Figure 7a), while parameters of $F_0$, $TR_0/RC$, and $ABS/RC$ were significantly lower than those of other treatments (Figure 7b). This indicated that the degree of heat damage of T2 was relatively weak under high-temperature conditions, and it explained that T2 could inhibit or alleviate the heat damage, maintain the relatively high photosynthesis ability of seedlings, and facilitate the growth and development of grapevine seedlings.

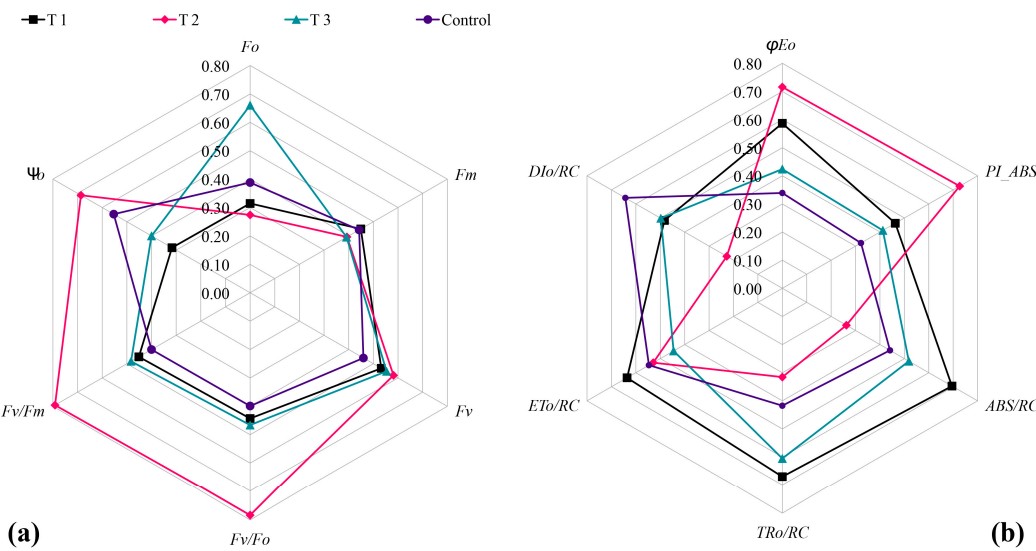

**Figure 7.** Comparison of Chl a fluorescence parameter in different treatments. (**a**) $F_0$, $F_v$, $F_m$, $F_v/F_0$, $F_v/F_m$, and $\Psi_0$; (**b**) $\varphi E_0$, $PI_{ABS}$, $ABS/RC$, $TR_0/RC$, $ET_0/RC$, and $DI_0/RC$.

## 4. Discussion

With the global climate changing, high-temperature conditions are becoming more frequent and lasting longer, significantly affecting plant growth, development, and crop yield. Grapevines, constantly exposed in the field, must withstand various unavoidable abiotic stresses during their growth and development. With the continuous increase in global temperatures, heat stress has gradually become a major limiting factor in the development of the grape industry [2,21]. Heat stress (>35 °C) can significantly impede the photosynthesis and nutrient metabolism of grapes, leading many high-quality varieties to lose their inherent characteristics. This directly affects the commodity and market value of grapes [3,22]. More importantly, dealing with heat stress will pose an unavoidable challenge for the global grape industry [15,23–25]. Turpan, Xinjiang, has become one of the most renowned grape-producing areas in China, which is attributed to its superior light and heat conditions that provide a unique environment for viticulture growth. However, the distinctive geographical condition also results in an atmosphere where temperatures equal or exceed 35 °C for over 100 days a year. The prolonged high-temperature conditions can affect the photosynthetic process, limiting the synthesis and transfer of photosynthetic products and significantly impacting its growth and development. When heat stress surpasses the plant's regulatory capacity, it manifests as injury symptoms in the phenotype, ultimately leading to wilting and, in severe cases, the death of the plants [5,26,27].

The 'Thompson Seedless' cultivar plays a pivotal role in Turpan, covering a planting area of 32,640 hectares, which accounts for 90.03% of the viticulture in the region [28]. High temperatures have frequently impacted the photosynthesis of plants, limited grape production, and impeded the growth of grapevine seedlings. This significantly affects the overall growth, development, and yield of grapevines. Biostimulants have been widely employed in crops due to their safety, non-polluting nature, low residue levels, and their ability to facilitate the growth and development of crops [29–31]. However, it remains uncertain whether they effectively work on grapevine seedlings under naturally high temperatures. Simulating high temperatures indoors may not fully and accurately reflect the growth of seedlings, particularly when considering new production methods [32–34]. Therefore, we planned to analyze the effects of various biostimulants on the growth and development of grape seedlings, as well as the fluorescence characteristics of chlorophyll, under natural high temperatures in Turpan. This study may not only offer insights into enhancing the cultivation of grapevine seedlings but also explore methods for cultivation in high-temperature conditions. Additionally, it holds reference significance for grape production in thermal and arid areas.

High temperature has a deleterious impact on plant osmotic adjustment by increasing evapotranspiration, which has an irreversible impact on solute generation, which is required for stress tolerance. Grapevines are regarded as a model perennial fruit crop for heat tolerance research, revealing that high temperatures pose various challenges to the development of high-quality grape berries [35]. We found that all biostimulants have diverse effects on the growth and development parameters of grapevine seedlings. The grapevine seedlings treated with biostimulants exhibited better performance than those in the control group in terms of plant height, stem diameter, and root development. This indicates that all three biostimulants have a positive effect on the growth of grapevine seedlings, which is consistent with findings from previous studies [36,37]. Moreover, the dry leaf weight, fresh leaf weight, leaf area, and chlorophyll content of each treatment were higher than those of the control. This indicates that the biostimulants also had a positive effect on leaf growth, consistent with findings from previous studies on *Pyrus* [38], *Gossypium* [39], *Maize* [40], and many other crops.

We also observed that as the duration of high temperature prolonged, the control group exhibited lower values compared to each treatment in features such as $F_v/F_m$, $F_v/F_0$, $F_v$, $\varphi E_0$, and $PI_{ABS}$. In contrast, $DI_0/RC$ was significantly higher in the control group than in any of the treated groups. Among the biostimulants, T2 was significantly higher than that of others treated in aspects of $\Psi_0$, $F_v/F_m$, $F_v/F_0$, $\varphi E_0$, and $PI_{ABS}$ (Figure 7a), while parameters of $F_0$, $TR_0/RC$, and $ABS/RC$ were significantly lower than those of other treatments. Some other parameters, like the dry and fresh weight of leaves, were comparatively higher in T2 when compared with any other treatments. The overall findings of various parameters showed that the degree of heat damage of T2 was relatively weak under high-temperature conditions. Overall findings indicate that the light energy conversion efficiency, activity of the PSII center, optical quantum yield, and photochemical properties of the control group were lower than those of each treated group, and the heat dissipation ratios in the control group were significantly higher than in all the treated groups during the high-temperature period. Furthermore, these suggest that three biostimulants had a positive effect on maintaining high levels of photosynthetic efficiency in grapevine seedlings under high temperatures [26]. Moreover, the results indicate that biostimulants could inhibit or alleviate damage to the photosynthesis of seedlings under high temperatures, thus enabling the maintenance of a relatively high level of photosynthesis. This conclusion is consistent with previous studies [41–44]. Furthermore, three biostimulants promoted the accumulation of chlorophyll or inhibited its decomposition in grapevine leaves. They assisted in maintaining a consistently high level of chlorophyll, facilitating the growth of grapevine seedlings during periods of high temperature. This indicates that biostimulants are not only beneficial to vine growth but also contribute to enhancing the heat tolerance of grapevine seedlings. And also, we should enhance our ability to comprehensively understand the various variables influencing heat stress and develop strategies for heat-tolerance grapevine breeding and cultivation [35,45].

## 5. Conclusions

Biostimulants play a crucial role in enhancing the growth and development of the roots, leaves, and overall health of vine seedlings. They contribute to promoting the accumulation of chlorophyll (Chl) and inhibiting its degradation in leaves. This ensures that photosynthetic efficiency remains at a high level, which is beneficial for the growth of grapevine seedlings, particularly under high-temperature conditions. Among these biostimulants, '6-B-2' (treated 2) stands out as particularly advantageous for the growth and development of vine seedlings in thermal and arid areas. It demonstrates the most effective results in alleviating heat stress and facilitating seedling growth.

**Author Contributions:** Conceptualization, J.W. and X.W.; methodology, R.A., J.W. and W.S.; software, J.W., C.Z., V.Y. and F.Z.; validation, J.W., H.Z. and X.W.; investigation, R.A., S.B. and Y.M.; writing—review and editing: J.W., H.Z. and X.W.; visualization, J.W. and V.Y.; supervision, H.Z. and X.W.; funding acquisition, J.W. and X.W. All authors have read and agreed to the published version of the manuscript.

**Funding:** This research was funded by the Natural Science Foundation of Xinjiang Uygur Autonomous Region (2023D01A96), Xinjiang Uygur Autonomous Region Tianshan Talents Training Program (2022TSYCJC0036), Xinjiang Uygur Autonomous Region Innovation Environment Construction Special Program (PT2314), and Xinjiang Uygur Autonomous Region Tianchi Talent—Special Expert Project (Xiping Wang, 2022), and the Hunan Innovative Province Construction Program (2023WK4008-4).

**Data Availability Statement:** The data that support the findings of this study are available upon request from the corresponding author.

**Acknowledgments:** We would like to thank the supportive and scientific staff in the research lab and grape research farm of Xinjiang Academy of Agriculture Sciences. Thanks to Wei Shi from the Turpan Eremophytes Botanical Garden, the Xinjiang Institute of Ecology and Geography, and the Chinese Academy of Sciences, who provided the biostimulant products and funding for testing.

**Conflicts of Interest:** The authors declare that they have no conflicts of interest.

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
