# Peer review of "Effects of Different Biostimulants on Growth and Development of Grapevine Seedlings under High-Temperature Stress"

_horticulturae, doi:10.3390/horticulturae10030269_

Round 1

Reviewer 1 Report

Comments and Suggestions for Authors

Dear authors, your article “Effects of different biostimulants on the growth and development of grape seedlings under high temperature stress” has some important weaknesses.

Considering the high heterozygosity of the vine, grape varieties are propagated by vegetative propagation, usually via cutting and grafting, while sexual propagation via seedlings is limited to genetic improvement. Therefore, it is not correct to study the effects of biostimulants on seedlings (each seedling is a different variety from each other), but cuttings should be used.

For the scientific community it is important to have information on the type of biostimulant, at least on the main component: without this information the research appears more suitable as additional information for the marketing of biostimulants than as a scientific article.

It is important to report statistics in each figure and table: statistical differences are often indicated without showing the results of the statistical analysis. Statics must be added in Figure 3a/b, Figure Figure 4b, Figure5, Figure 7.

Reviewer 2 Report

Comments and Suggestions for Authors

I read with interest the manuscript entitled “Effects of Different Biostimulants on Growth and Development of Grapevine Seedlings under High Temperatures Stress”. The aim is to provide a basis for reinforcing the cultivation of grape seedlings, offering a reference for alleviating heat stress, and exploring pathways for stress-resistant cultivation. The subject of the article is important and has great relevance for the scientific environment of the study area. Therefore, the manuscript needs some adjustments so that it can then be forwarded to the publication process. The manuscript has the potential for publication in the journal Horticulturae and needs the following adjustments:

ABSTRACT

- It is mentioned that chlorophyll fluorescence parameters were measured. Was it just that? Quote the rest of the analyses.

- Mention which biostimulants were used.

- More than the aforementioned photosynthesis was analyzed. To review.

- Review the objective. Standardize with what is mentioned in the Introduction. They must be similar.

- Replace repeated keywords in the title.

INTRODUCTION

- What would be biological stress? This is mentioned in the first paragraph.

- Review the second paragraph. It is mentioned about the variety chosen in the study. This information must be placed in the Material and Methods section.

- Standardize the objective. It should be similar to the one mentioned in the Abstract.

MATERIAL AND METHODS

- Were the biostimulants applied every 7 or 10 days? The application interval must be standardized. To correct.

- What is the composition of these biostimulants? How can this study be replicated without information about the products applied?

- Delete the first paragraph of subtopic 2.4. It is not necessary to mention the methodology used to take the photo.

- Were three leaves per plant used for chlorophyll analyses? From which part of the plant were they collected? Young or mature leaves? To review.

- Were 3 leaves randomly collected to measure leaf area? This method of collecting the number of leaves can harm or mask leaf area analyses.

- Why was only 15 minutes used with the leaves in the dark to determine chlorophyll fluorescence? The recommended time is 30 minutes.

- Delete the last part of the last paragraph. It is not necessary to mention that means and standard errors are presented in tables and figures. This is mandatory to be done in all jobs.

RESULTS AND DISCUSSION

- Cite the results and then cite the table or figure. This order is reversed. Review this throughout the text.

- In line 303 there is a possible justification for the result found. This is a discussion and should be relocated to the other section.

Reviewer 3 Report

Comments and Suggestions for Authors

The researchers evaluated phenotypes of grapevine seedlings to verify the effects of different biostimulants on the growth, development and photosynthesis of the seedlings. The study is interesting but presents serious problems that prevent its publication in the current version. There is a lot of data missing from the methodology. The authors need to prove that the cultivation of plants would not have stresses other than high temperature, for example nutritional. To achieve this, it is important to detail the amount of nutrients provided to compose the substrate and, more importantly, include a chemical analysis of the nutrient levels available in the cultivation medium used in the experiment. The vine seedlings were obtained at 1 year of age but the authors did not indicate the height, stem diameter and the respective variation of these indicators in the population used for the experiment. The concern here would be to indicate that the seedlings were homogeneous before starting the experiment, hence these variables are important. The critical point of the research is the lack of description of the composition of the biostimulants '24-B-2' (T1), '6-B-2' (T2) and 'I59' (T3). Without this, the research loses its scientific basis because it is not possible to understand why a certain biostimulant (6-B-2) resulted in an improvement in the development of seedlings and why the other biostimulants had lower performance. Therefore, it is not possible to have a discussion or a defense of the results obtained in the research, but only a description. Which amino acid or organic compound was responsible for the effects obtained in the plant? How to carry out research "in the dark" just indicating that one product is better than another. This is not science but technical testing. As presented, the research would only be descriptive indicating the effects of treatments on physiological parameters but it is not possible to understand the mechanisms involved that explain the results obtained and therefore scientific advances are limited. We understand that this is a product with a patent, but the main compounds must be indicated to result in scientific work and not a technical report.

Round 2

Reviewer 1 Report

Comments and Suggestions for Authors

Most of my suggestions were carried out.

Author Response

Thank you for your kind comments.

Reviewer 2 Report

Comments and Suggestions for Authors

Dear,

The manuscript was corrected according to the suggestions proposed in the previous version. Therefore, the article has the potential to be published in this journal.

Author Response

Thank you for your kind comments.

Reviewer 3 Report

Comments and Suggestions for Authors

The researchers evaluated the effects of several biostimulants on the growth and development of grape seedlings under high temperatures.

There has been improvement in the text but it is important to move forward with improvement.

The introduction does not clearly indicate the state of the art and the novelty of the research.

The authors need to discuss the potential of biostimulants to minimize cellular oxidative stress.

To deepen the discussion on the factors that most contributed to the recommendation of the best biostimulant, it is necessary to include a multivariate analysis that will improve the quality of the research.

The authors indicated the maximum and minimum temperature, but this information is very specific. It is important to include the average value of all maximum temperatures. and minimum temperatures. These temperatures are biologically important for understanding the level of stress.

It is very important for the authors to describe in detail the foliar spraying of biostimulants. What time of day was the spraying carried out? At this point include the ambient temperature. Include data on the sprayer used, such as nozzle size and spray flow. What phenological stage of the plant was the biostimulant applied to? Was a surfactant/spreader added to the solution you used for foliar spraying? Which?

And the amount of solution applied per plant. Without this data, it is not possible to assess whether the biostimulants were actually well absorbed by the leaves.

The discussion needs to be improved to defend the best treatment/biostimulant. You would have to join figures 4 and 5.

Author Response

Thank you for your comments on our revised manuscript, we have revised the manuscript as per recommendation. Please see the attachment.
